# How Does Neural Network Model Capacity Affect Photovoltaic Power Prediction? A Study Case

**DOI:** 10.3390/s23031357

**Published:** 2023-01-25

**Authors:** Carlos Henrique Torres de Andrade, Gustavo Costa Gomes de Melo, Tiago Figueiredo Vieira, Ícaro Bezzera Queiroz de Araújo, Allan de Medeiros Martins, Igor Cavalcante Torres, Davi Bibiano Brito, Alana Kelly Xavier Santos

**Affiliations:** 1Computing Institute, A. C. Simões Campus, Federal University of Alagoas—UFAL, Maceió 57072-970, Brazil; 2Center of Agrarian Sciences, Engineering and Agricultural Sciences Campus, Federal University of Alagoas—UFAL, Rio Largo 57100-000, Brazil; 3Electrical Engineering Department, Center of Technology, Federal University of Rio Grande do Norte—UFRN, Natal 59072-970, Brazil

**Keywords:** photovoltaic solar energy, short term energy forecast, multilayer perceptron, recurrent neural network, long short-term memory

## Abstract

The use of models capable of forecasting the production of photovoltaic (PV) energy is essential to guarantee the best possible integration of this energy source into traditional distribution grids. Long Short-Term Memory networks (LSTMs) are commonly used for this purpose, but their use may not be the better option due to their great computational complexity and slower inference and training time. Thus, in this work, we seek to evaluate the use of neural networks MLPs (Multilayer Perceptron), Recurrent Neural Networks (RNNs), and LSTMs, for the forecast of 5 min of photovoltaic energy production. Each iteration of the predictions uses the last 120 min of data collected from the PV system (power, irradiation, and PV cell temperature), measured from 2019 to mid-2022 in Maceió (Brazil). In addition, Bayesian hyperparameters optimization was used to obtain the best of each model and compare them on an equal footing. Results showed that the MLP performs satisfactorily, requiring much less time to train and forecast, indicating that they can be a better option when dealing with a very short-term forecast in specific contexts, for example, in systems with little computational resources.

## 1. Introduction

The natural depletion of fossil fuels worldwide boosts a search for alternative energy sources to meet increasing demands. In this context, new research and investments aimed at improving renewable energy efficiency are increasing. Among all the options, photovoltaic energy (PV) stands out as a clean, inexhaustible, and environmentally friendly power source. Worldwide, the contribution of renewable energy sources grew by 200 gigawatts in 2021, with photovoltaic energy accounting for 55.7%, despite aftershocks from the pandemic and a rise in global commodity prices [1]. This trend is no different in Brazil, with renewable sources representing 83.9% of the installed generation capacity in the Brazilian energy generation matrix in April 2022 (hydraulic, biomass, wind, and photovoltaic), with photovoltaic energy responsible for 7.9%. In addition, PV showed an installed capacity growth, between April 2021 and April 2022, of 72.6%, reiterating its growing importance in energy generation in Brazil [2].

However, the electrical energy generated by a photovoltaic panel depends on environmental factors such as solar irradiation and the temperature of the photovoltaic cell, which makes its production inherently variable and uncertain. This stochastic characteristic of photovoltaic production, at a high penetration level, can bring instability to the traditional electrical grid due to voltage fluctuations and generated power, resulting in the deterioration of the grid’s operating frequency [3]. Hence, forecasting models are essential to ensure a stable and economically advantageous integration from photovoltaic energy sources to traditional distribution networks [4].

According to Refs. [5,6,7], prediction models contribute to better planning, distribution, and storage of generated energy. In addition, they can assist in several other technical aspects related to the better use of photovoltaic systems, such as in the prevention of overvoltage, attenuation of ramps, and execution of control actions [8,9,10]. In this context, many studies are being developed to create PV energy generation forecasting models with increasing accuracy and lower computational cost.

Variable forecasting is already widespread in the literature, where computer models seek to forecast the most diverse variables, from house prices to water quality conditions [11,12]. Regarding photovoltaic energy production, Ref. [13] classify the forecast models into four methods: statistical, artificial intelligence, physical, and hybrid. Statistical approaches are based on formulations guided by historical measurement data (time series) to make predictions, as shown by [14], where autoregressive models are applied. In the Artificial Intelligence method, advanced AI techniques are used, along with time series, to make predictions, as in support vector machines and random forests [15]. Physical models use detailed models of photovoltaic modules together with numerical weather forecasts or satellite images [16]. Finally, the hybrid approach combines two or more previous methods [17,18].

Statistical and artificial intelligence methods are said to be data-driven, where previous operation data is needed to train/calibrate forecast models. Thus, the greater the quantity and quality of existing data, the greater the tendency to produce good forecast models. In comparison, the AI method usually presents the best results, but they demand a more significant computational effort. As the name implies, the physical method is based on the physical characteristics that influence PV generation, such as PV system metadata and meteorological data. Therefore, the great advantage over data-driven models is that physical models can be implemented even before the photovoltaic system is in operation, although they generally have lower accuracy [19].

The work carried out by [5] highlights the artificial intelligence approach, especially in using artificial neural networks (ANNs), presenting excellent performance compared to the other methods, even in rapidly changing environmental conditions. One of the simplest ANN is the MultiLayer Perceptron (MLP), whose efficiency is demonstrated by [20,21] for the prediction of solar irradiance and by [22] for short-term prediction of the electrical energy generated by a photovoltaic module. Another class of ANN that has been gaining much attention is the recurrent neural network (RNN), more specifically the so-called Long Short-Term Memory (LSTM), which can be attested by: [10,23,24].

Often works have tackled the problem of PV power generation using sophisticated recurrent neural networks (RNN) for accurate prediction [25,26,27,28,29]. However, the computational cost of a deep recurrent neural network with many parameters can be prohibitively expensive and take too long to train [30]. Thus, given the importance of predictive models of photovoltaic power generation, this work aims to answer the following question:How complex does the network architecture need to be to predict very short-term power production demands satisfactorily?

In order to answer the research question, we compared the performances of MLP, fully connected RNN, and LSTM to predict five minutes of power generation using variables of the past 120 min, as [31] indicate that horizon presents the best results for very short-term predictions. The hyperparameters were fine-tuned to ensure we had the optimal (minimum) error for each architecture. Furthermore, the results were compared between each other and to a naive baseline, showing no significant reduction in error levels for PV power generation prediction provided by models with recurrence. Therefore, we conclude that lighter MLP is sufficient to perform the prediction satisfactorily.

This work presents the following sections: Section 1, the introduction is shown. Section 2 presents the methodology and details the dataset used, and the results obtained are exposed and discussed in Section 3. Finally, in Section 4, the conclusions about the work are presented.

## 2. Materials and Methods

### 2.1. Database Acquisition

The instrumentation stage included installing the following sensors: solar irradiation, ambient temperature sensor, and operational temperature of the PV module and power injected into the electrical grid, in addition to other transducers. Figure 1 presents the real experimental setup, where all the transducer elements are visualized together with the data acquisition unit, the CR1000 datalogger from Campbell Scientific.

Each sensor was connected to the datalogger, equipment responsible for collecting and storing data, the selected equipment is the Scientific Campbell CR1000. The equipment was installed with conventional power and backup power for alternative supply. The measurement campaign of this research began on 1 January 2019, continuing continuously until April 2022, resulting in approximately 1200 monitored days. The temporal cadence of storage for each variable was predefined in 1-min averages; so, for 1 day, there are 1440 records per minute for each variable, so just over 1.5 million records were stored per variable. Figure 2 illustrates the experimental PV system with a total capacity of 5 kW under STC and the main points used in data acquisition to obtain the variables of interest.

### 2.2. Database Analysis

Initially, a thorough analysis of the database was carried out. Figure 3 shows the correlation between the seven variables analyzed. The variables most related to the generated power are solar irradiation and the temperature of the photovoltaic module. By means of that, we selected three variables to make the PV power forecasting: irradiation, the temperature of the photovoltaic module, and the power itself.

Table 1 shows the main characteristics of the instant power data set, which is our target variable. One critical factor of this data set is that it has a reasonable amount of outliers, as illustrated in Figure 4a, especially the values above 5 kW, which is the limit of the photovoltaic system used in this work. This fact is most likely due to sensor measurement errors.

In addition, we sought to observe the photovoltaic energy distribution of the database in hours, days, and months. As shown in Figure 5a, the maximum PV power is generated at noon, presenting operating hours of 7 am to 6 pm. The distribution by day shows small but not significant fluctuations (Figure 5b). Finally, the distribution by month, Figure 5c, showed a lower production in the rainy months of Maceió, mainly May, June, and July, as expected.

### 2.3. Data Preprocessing

After analyzing the available data, it was observed that some instances had missing or negative values due to technical problems and sensor noise. Thus, preprocessing was necessary, where we removed the missing values and truncated the negatives to zero. Outliers associated with measurements performed at night (moments with zero solar irradiation and consequently without energy generation) were left in the database because the models must operate under such conditions in the day-to-day.

We separated the selected three variables that we analyzed from the correlation matrix and performed the normalization process in values between 0 and 1. Then, to form the input sets of the models, we grouped 120 previous values of each sample in sequence for both the PV power, the irradiance, and the temperature of the photovoltaic cell. For output sets of the models, we grouped the five following values in sequence for each PV power sample.

Afterward, the database was divided, reserving the last 20% of the data for the test phase and the rest for training and validation. In the end, the test set makes it possible to view model performance on data that was not used during the training and validation process, mirroring the neural network’s performance in an actual operation scenario.

### 2.4. Prediction Models

For the implementation of all neural networks in this work, the programming language Python 3.8 was used, as well as the open-source libraries Keras [32] and scikit-learn [33], as they facilitate the implementation of well-performing forecasting models.

The first models created were the three baseline models. In the first one, the “average” baseline, the forecast is given by the average of the variable’s value of the previous five days at the same time. In the second, the “one-minute” baseline, the idea is to use the variable’s value from the previous minute. The last one is a simple linear regression model, a machine learning algorithm widely used for predicting variables due to its simplicity.

Three architectures were used for the development of neural networks, namely: MLP, RNN (fully connected), and LSTM. Each of these models receives as input three vectors containing values of power, irradiance, and temperature of the PV module, in 120 consecutive time steps (2 h) and outputs a vector with the following five-time steps through the output layer (Dense layer). The main structural differences between the models were in their hidden layers, where each has specific layers of its type and varies in the amount of them. Figure 6 describes the general structure of the created models.

Initially, the choice of the number of hidden layers and how many cells would compose them was carried out empirically through several experiments. Subsequently, we used Keras Tuner for a fairer comparison. This framework helps to choose the ideal set of hyperparameters for a neural network, finding a final model close to the optimal one for each architecture. Hyperparameter optimization is a significant problem in Deep Learning due to its overfitting tendency as a consequence of the high capacity provided by the use of many hidden layers. In Keras Tuner, we chose to use Bayesian Optimization due to its good result in less search time than, for example, Grid Search [34]. In the optimization process, we used the following configuration:
Number of hidden layers between 1 to 4;
–Neurons from 32 to 256 with 32 increase at each step;–Rectified Linear Unit (ReLU) for the MLPs and Tangent Hyperbolic (Tanh) for the RNNs and LSTMs, as hidden layers activation function;Dropout layer after each hidden layer;
–Dropout rate from 0 to 0.8 with 0.2 increase at each step.5 Neurons in the output layer;Learning rate of $1\times 10^{-3}$;
–Early stopping;–Reduce on plateau;Linear as output layer activation function.

As shown, the search space between all neural network topologies remained the same, even with the much more significant time required for the recurrent network tuning process, especially in the case of LSTMs.

### 2.5. Training and Testing

During the training, we used the backpropagation algorithm for the MLPs and the backpropagation through time (BPTT) algorithm for the RNNs and LSTMs. In addition, Adam was used as the optimizer as it is well-suited to a wide range of non-convex optimization problems in machine learning. Ref. [35], the MAE was the loss function, and the batch size was 512. Two stopping criteria were used to prevent overfitting: when training reaches 128 epochs or when the loss function has an improvement smaller than $10^{-9}$ for twenty consecutive epochs (Early Stopping).

The validation process occurs during training at the end of each epoch by making predictions using 20% of the data taken randomly from the training set, excluding them from training. This stage is crucial because it provides the value of the loss function that makes it possible to use the model checkpoint, which saves the model in the moments when it reaches the best performance. In addition, the validation values allow using the Reduce on Plateau technique, which decreases the learning rate when the validation metric stops improving for longer than the patience number of epochs allows.

Lastly, the performance of each model was evaluated on the test set to assess the model’s generalization capabilities for forecasting unseen data. The statistical metrics used to evaluate the forecast models developed in this work were: Mean Absolute Error (MAE) (Equation (Equation 1)), Root Mean Squared Error (RMSE) (Equation (Equation 2)), and Standard Deviation of the Error (STD) (Equation (Equation 3)).
(1)$$MAE=\frac{1}{N}\left(\sum_{i=1}^{N}\left|y_{i}-\hat{y_{i}}\right|\right)$$
(2)$$RMSE=\sqrt{\frac{1}{N}\sum_{i=1}^{N}{\left(y_{i}-\hat{y_{i}}\right)}^{2}}$$
(3)$$STD=\sqrt{\frac{1}{N-1}\sum_{i=1}^{N}{\left(e_{i}-\bar{e}\right)}^{2}}$$
where *N* is the number of data samples, $y_{i}$ is the *i*-th instant power generated by the PV module, $\hat{y_{i}}$ is the *i*-th instant PV power forecast by a model, and $e_{i}$ is the *i*-th difference between $y_{i}$ and $\hat{y_{i}}$.

MAE measures the average magnitude of the errors in a set of predictions without considering their direction. Similarly, RMSE expresses average error, but as the errors are squared before they are averaged, the RMSE gives a higher weight to significant errors and outliers. The standard deviation evaluates the variation from the average of the forecast errors.

In addition, neural network models were also evaluated for their complexity and usability through the following metrics: number of trainable parameters (NTP), time of one epoch of training (TT), and inference time (IT). The NTP, as the name describes, is the number of parameters that will be calculated during the training process of the artificial neural networks, estimating how complex the network is. TT is the average time it takes to complete one epoch in training. Finally, IT represents the time it takes a model to produce its output. In the context of this work, to ensure that it is calculated correctly, the inference time consists of the average time spent to predict one hundred random samples.

## 3. Results and Discussion

As discussed in the previous section, several configurations were considered for the MLP, RNN, and LSTM networks. Thus, given many evaluated models, each type’s best-performing one hidden layer models were chosen to compare with the baselines and models found by the hyperparameter optimizer. The best configuration found empirically for the one-layer model were 60 neurons, 120 and 256 neurons for the MLP, RNN, and LSTM networks, all with a learning rate of $10^{-3}$. In the case of networks found by Keras Tuner, the final configuration found for MLP contains four hidden layers of 32, 512, 512, and 512 neurons, respectively. In comparison, the final configuration found for the simple RNN contains two hidden layers of 32 neurons. At the same time, the LSTM presented two hidden layers of 256 neurons. Finally, it is important to note that all models used did not have any dropout layer.

The performance of each model can be observed in Table 2. One can see that the optimized neural network models are better than the baselines. The optimized LSTM, for example, showed an improvement of approximately 63.791% in MAE and 45.184% in RMSE over the “Average” baseline. It also presented an improvement of approximately 10.803% in the MAE and 10.07% in the RMSE over the “One-minute” baseline. In comparison with the Linear Regression model, we observed an improvement. However, it is smaller than what was seen in the previous ones, showing an improvement of approximately 21.488% in MAE and approximately 1% in RMSE, which is something expected in a comparison between machine learning models.

Among neural network models, the results are very similar. The RNN and LSTM networks optimized by Keras Tuner showed the best metric values in the test phase, closely followed by the optimized MLP. However, the one-layer neural network models presented results closer to the best baseline, showing the importance of a good choice of hyperparameters.

Given the comparable performance of all optimized ANNs, it is crucial to analyze their usability and complexity. Table 3 presents the TT, NTP, IF, and MAE for each neural network model on the same computer equipped with an Nvidia gtx 1660 super graphics card and an Intel i7-10700 desktop processor. One can see that the inference time is close between all models, making the most significant difference in the training time. The optimized LSTM, despite one of the best metrics, is the one with the longest training time and the most significant number of trainable parameters. On the other hand, the optimized MLP presented an excellent training time, about 319 s of difference for each epoch, with a very close MAE. Finally, the optimized RNN presents a middle ground between the two, a faster training time than the LSTM but still much longer than the MLP.

About LSTM, we observed a very similar performance to the other RNN, with the disadvantage of being a more complex network and having more extended training. Since the great advantage of LSTM networks is their ability to learn extended temporal contexts, in our prediction model, which only uses one hundred and twenty minutes elapsed, this advantage may not be well explored.

Then, to ensure that the prediction enhancements that the ANNs models provide are statistically significantly different from the baseline models, *t*-tests [36] of the MAE were performed using the test dataset. The result of the optimized LSTM and the “One-minute” baseline can be observed in Table 4, showing that the *p*-value is less than 0.05. Consequently, we can deny the hypothesis that the mean of the model’s MAE is equal to the baseline. In other words, the forecast results made by the LSTM model and the baseline model are significantly different. Furthermore, all the tests from the other models gave out similar results.

In the analysis of the forecasts of each minute, it was noticed that the MAE grows the farther away the predicted minute is, as can be seen in Figure 7, which shows the evaluative metrics of every minute of the optimized MLP. An MAE increase of 54.56% between the first and fifth minute can be observed, in addition to a significant increase in the STD as the forecast horizon increases, demonstrating that the error dispersion around the mean grows.

Finally, even with all the reservations made above, it can be said that the models produced good results, presenting a small error if we consider the power scale of our database, which has values that reach up to seven thousand watts. To illustrate this, we predicted three different days of the test dataset using the optimized LSTM model, one being a sunny day, another a partially cloudy day, and finally a cloudy one. The series represented by the red line is the power values generated by the photovoltaic panels. In contrast, the series represented by the blue dotted line comprises the values predicted by the network. Visually, the prediction on sunny days is almost perfect for both the first and the fifth minutes (Figure 8). In the case of partially cloudy and cloudy days, displayed in Figure 9 and Figure 10, we can observe the excellent result of the forecast of the first minutes, even if there are failures in some points. Regarding the fifth minute, the model encounters some more difficulties, especially in moments of sudden change, but generally, it is satisfactory.

## 4. Conclusions

The present work compares MLP, fully connected RNN, and LSTM models, for the very short-term prediction of electric energy generation by a photovoltaic system, through the time series approach. The presented results showed the importance of hyperparameter tuning for better model performance. Furthermore, the optimized models presented similar achievements, with very acceptable mean absolute error values, given the scale of the worked data. Thus, the study demonstrated that the models could predict a future five-minute power horizon using the PV power, irradiation, and PV module temperature values of two past hours, even with an increase in error as the forecast horizon increases.

Discussing the error metrics purely, the model that presented the best result were the RNN and LSTM models, which was expected, given their favorable characteristics for time series. However, the MLP models returned results very close to the best. Considering the usability and complexity of the models, we see an advantage of the MLP model over the others, as it takes a considerably shorter time to train. In addition, the study made it clear that LSTM was underutilized in the context of very short-term forecasting. Thus, there is no need for very complex architectures for very short-term production forecasts, especially in specific contexts where training time is essential, e.g., for future retraining.

As a continuation of the work, we intend to observe the underutilization of the LSTM, seeking to make predictions using a more extensive temporal context than what was used. In addition, new models for comparison need to be developed, from machine learning algorithms like SVR [37] and Random Forest [38], for the baseline, to other ANNs such as GRU network [39] and the Transformers [40], and those models could then be applied in the solarimetric station to observe how they behave in practice.

## Figures and Tables

**Figure 1 sensors-23-01357-f001:**
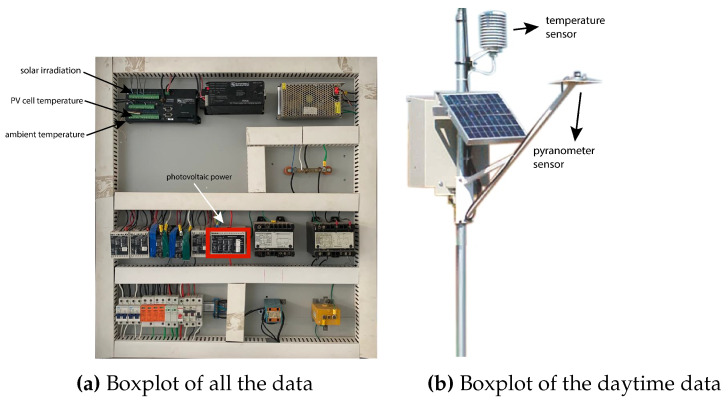
Data acquisition infrastructure.(**a**) Instrumentation panel and data acquisition unit of the photovoltaic system. (**b**) Climatological station of the experimental photovoltaic system.

**Figure 2 sensors-23-01357-f002:**
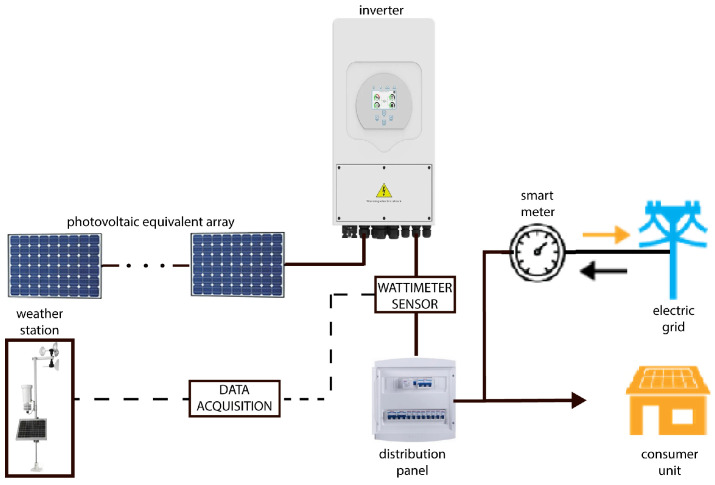
Illustrative diagram of the experimental PV system.

**Figure 3 sensors-23-01357-f003:**
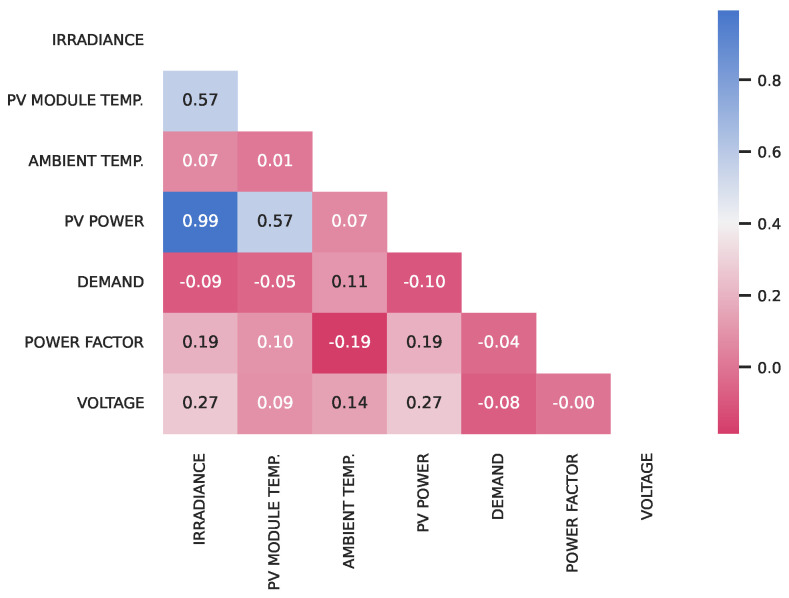
Correlation matrix between the variables in the database.

**Figure 4 sensors-23-01357-f004:**
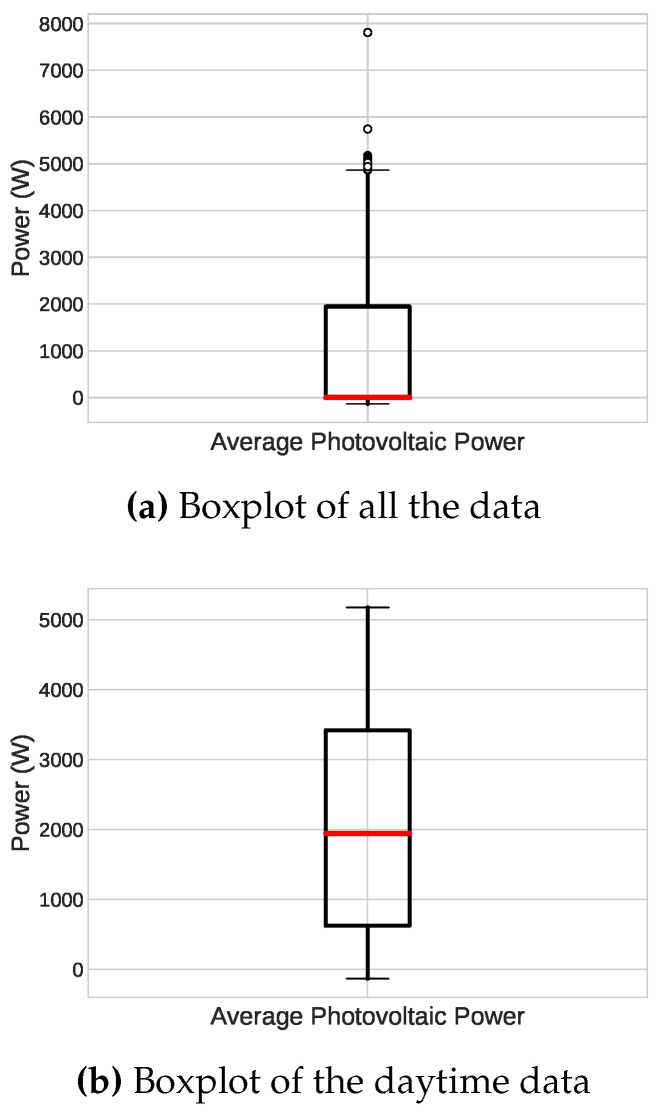
Boxplot of the instant power data. It is noticeable that the median is close to zero due to many samples acquired during the night. (**a**) all the data and (**b**) daytime data only.

**Figure 5 sensors-23-01357-f005:**
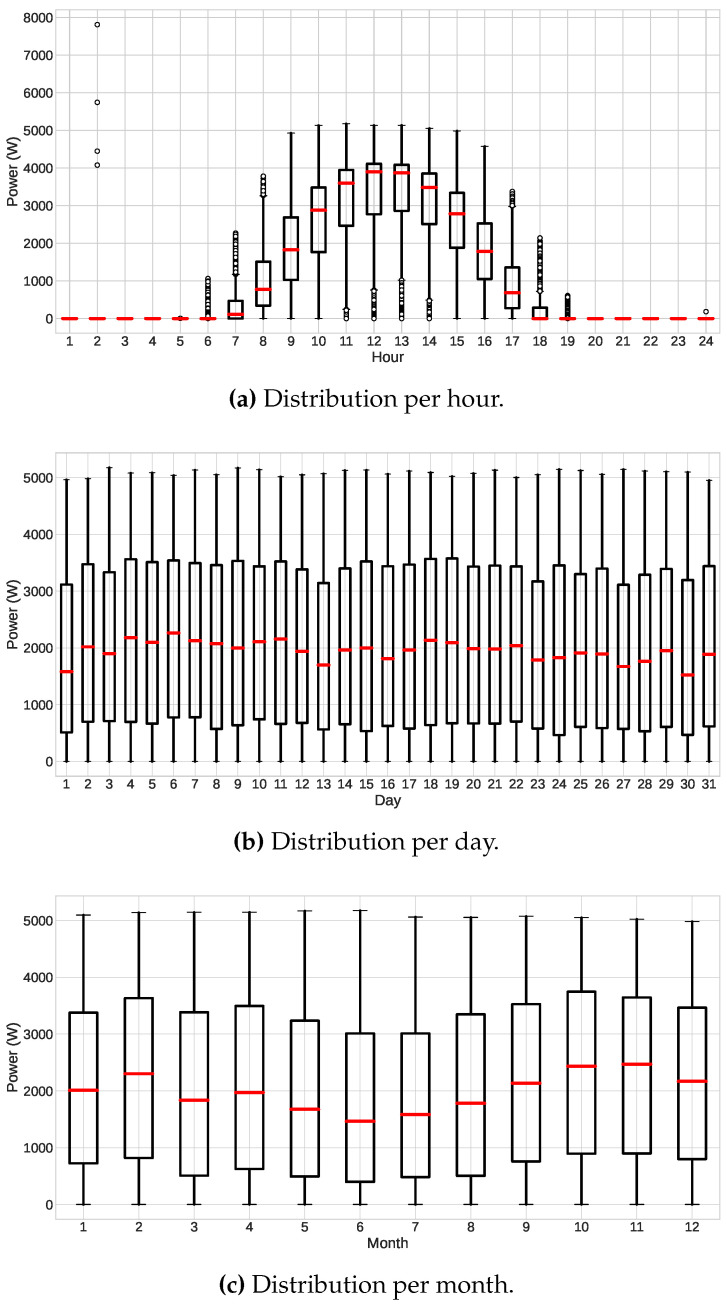
Distribution of the photovoltaic power values from the database, considering (**a**) the distribution in hours, (**b**) the distribution in days (daytime values only), and (**c**) the distribution in months (daytime values only).

**Figure 6 sensors-23-01357-f006:**
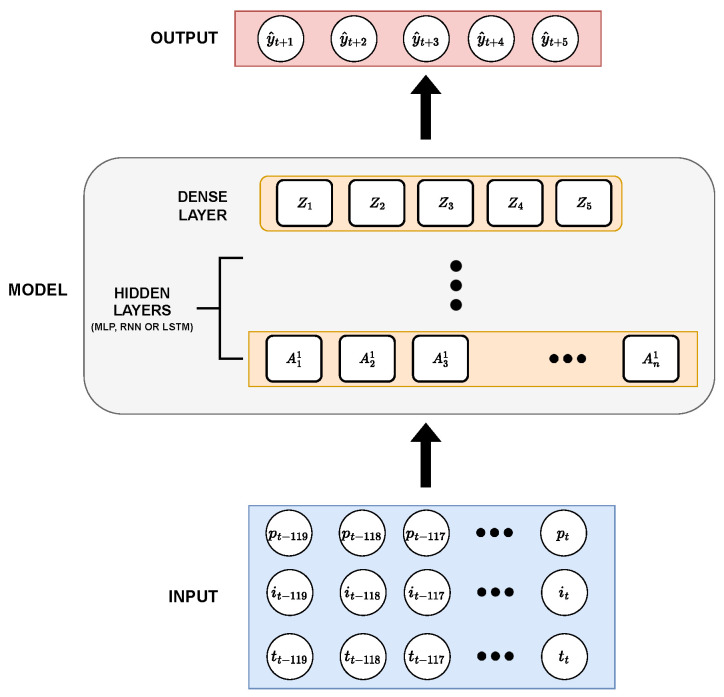
General neural network architecture developed for the present work.

**Figure 7 sensors-23-01357-f007:**
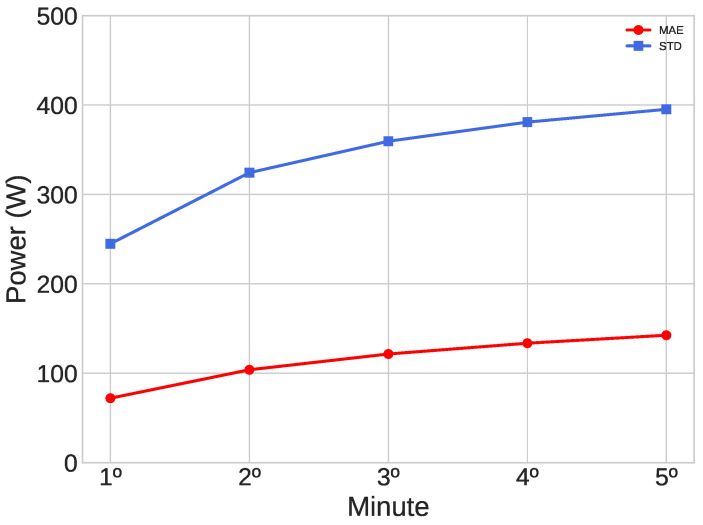
Mean Absolute Error (MAE) and Standard Deviation (STD) in the test phase of the optimized MLP model as a function of the minute prediction value.

**Figure 8 sensors-23-01357-f008:**
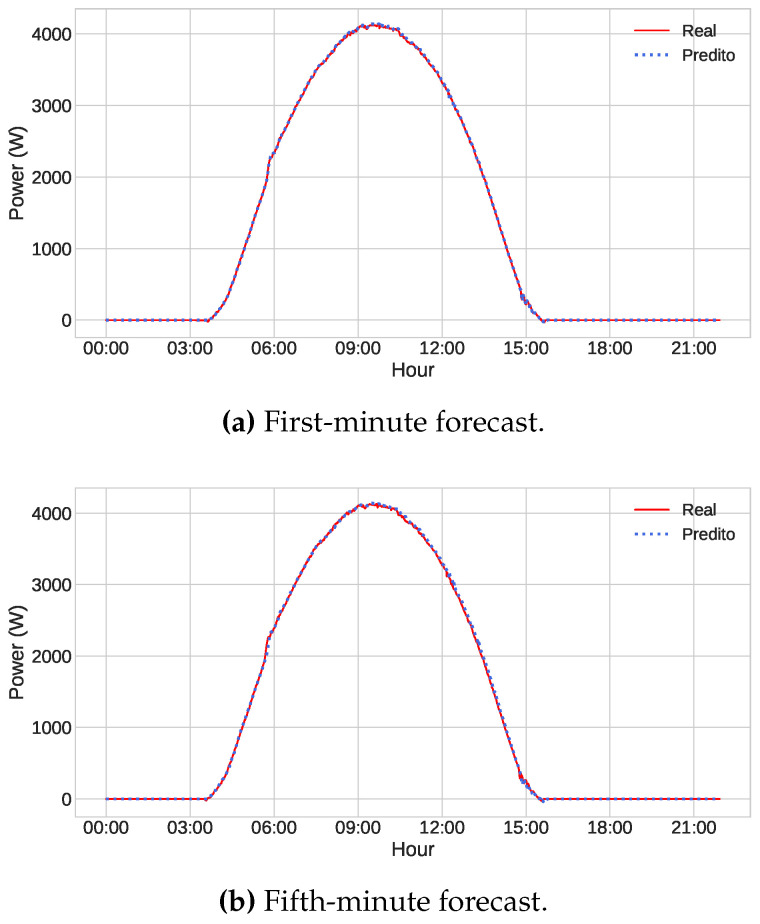
Power forecast by the optimized LSTM model (blue dots) and the actual power values (red line) on the sunny day of 02/07/2022, considering (**a**) the first-minute forecast and (**b**) the fifth-minute forecast.

**Figure 9 sensors-23-01357-f009:**
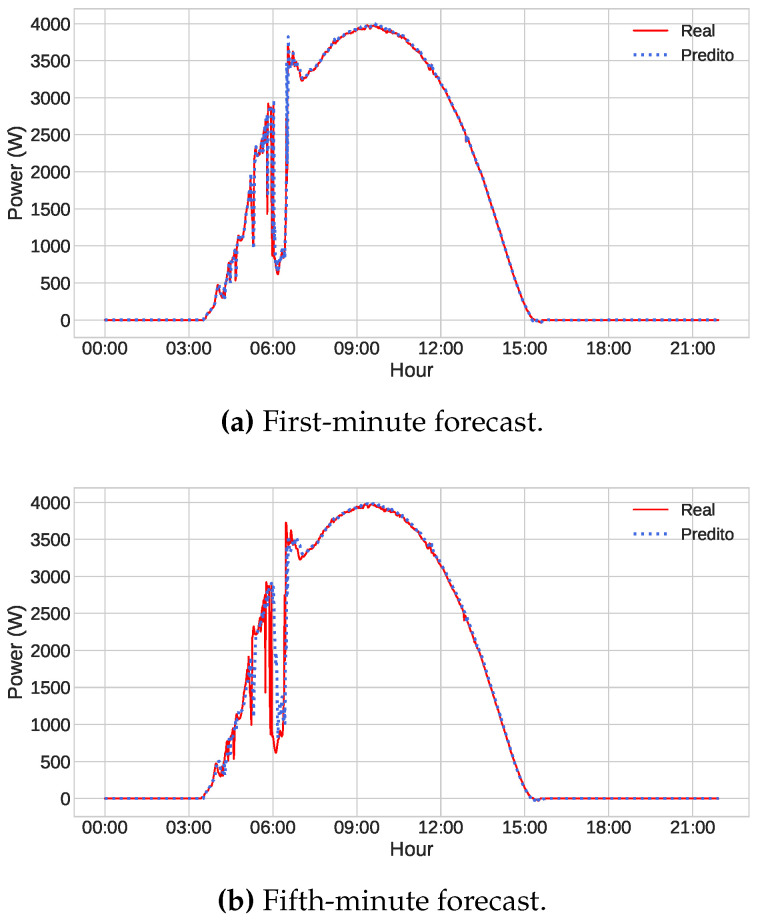
Power forecast by the optimized LSTM model (blue dots) and the actual power values (red line) on the partially cloudy day of 01/22/2022, considering (**a**) the first-minute forecast and (**b**) the fifth-minute forecast.

**Figure 10 sensors-23-01357-f010:**
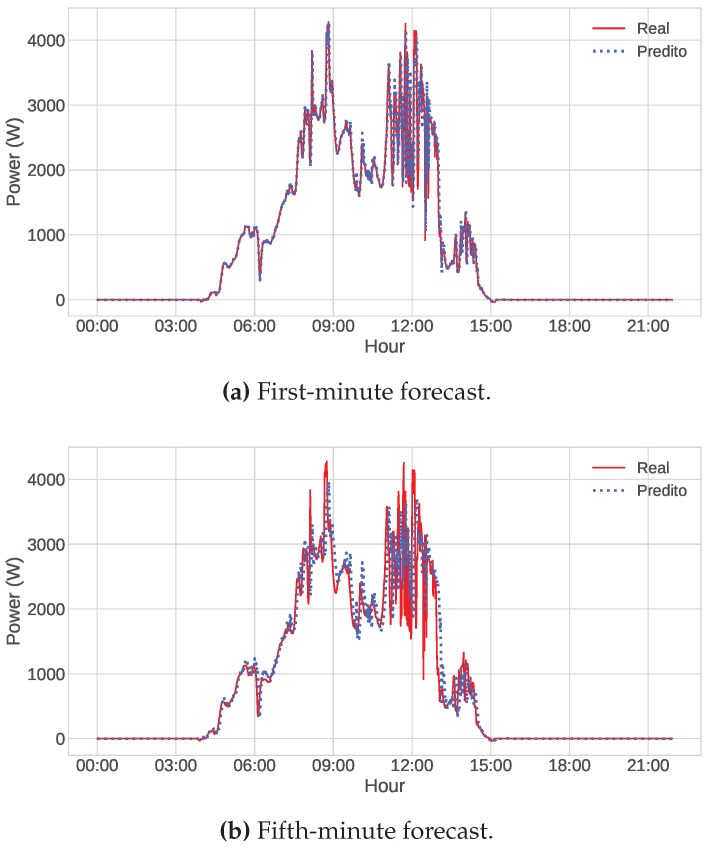
Power forecast by the optimized LSTM model (blue dots) and the actual power values (red line) on the cloudy day of 01/04/2022, considering (**a**) the first-minute forecast and (**b**) the fifth-minute forecast.

**Table 1 sensors-23-01357-t001:** Main characteristics of the target variable.

	Characteristics			
	Maximum	Minimum	Average	Standard Deviation
value	7807 W	−110.1 W	1006.517 W	1437.447 W

**Table 2 sensors-23-01357-t002:** Mean Absolute Error (MAE), Rooted Mean Squared Error (RMSE), and Standard Deviation (STD) measured (Watts) in the Training, Validation, and Test phase of each model.

	MAE	RMSE	STD
“Average” baseline	315.191 W	623.784 W	623.778 W
“One-minute” baseline	127.951 W	380.220 W	380.220 W
Linear Regression	145.364 W	345.226 W	345.212 W
One Hidden Layer MLP	125.113 W	350.306 W	349.889 W
MLP optimized	117.308 W	343.702 W	342.861 W
One hidden layer simple RNN	131.823 W	335.315 W	335.033 W
Simple RNN optimized	114.737 W	340.167 W	339.298 W
One hidden layer LSTM	128.482 W	330.753 W	330.545 W
LSTM optimized	114.728 W	341.933 W	340.993 W

**Table 3 sensors-23-01357-t003:** Time of One Epoch of Training (TT), Number of Trainable Parameters (NTP), Inference Time (IF), and Mean Absolute Error (MAE) of each neural network model.

	TT	NTP	IF	MAE
MLP	3 s	21,965	2.837 ms	125.113 W
MLP Optimized	4 s	556,325	2.887 ms	117.308 W
RNN	48 s	15,485	3.465 ms	131.823 W
RNN Optimized	125 s	3397	4.067 ms	114.737 W
LSTM	129 s	60,165	2.951 ms	137.235 W
LSTM Optimized	323 s	792,837	3.231 ms	114.728 W

**Table 4 sensors-23-01357-t004:** Results of *t*-test of the MAE from the optimized MLP and the baseline model.

	*t* Test			
	*t* Value	*p* Value	Tail	Degree of Freedom
value	16.980	≤0.05	Two-sided	577,676

## Data Availability

The database used in this work can be found, under request, at https://drive.google.com/file/d/1o_h3Kc-FdNPv11Tg10VC4CSNixCFDWPM/view?usp=share_link (accessed on 12 January 2023). The trained neural network and the normalizers can be found at https://drive.google.com/file/d/18GBEGzRjh_b6ASN_o4r5ZiLYbm9oR8ZR/view?usp=sharing (accessed on 12 January 2023). All of the software codes can be found in the Solar Power Forecasting repository at https://github.com/edge-softex/solar_power_forecasting (accessed on 12 January 2023).

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
