# Peer review of "How Does Neural Network Model Capacity Affect Photovoltaic Power Prediction? A Study Case"

_sensors, 2023, doi:10.3390/s23031357_

Round 1

Reviewer 1 Report

This paper compared the performance of MLP (Multilayer Perceptron), Recurrent Neural Networks (RNNs), and LSTMs for the task mentioned above. Each iteration of the forecast uses the last 120 minutes of data collected from the photovoltaic (PV) system (power, irradiation, and PV cell temperature), measured throughout 2019 to mid-2022 in the city of Maceió (Brazil), to predict the next 5-minutes of PV instant power. Hyperparameters were fine-tuned using an optimization approach based on Bayesian inference to promote a fair comparison. Results showed that the MLP has satisfactory performance, requiring much less time to train and forecast. However, if well-tuned, the recurrent neural networks still present the best results. Such results indicate that MLP models can be a better option when dealing with a short-term forecast in specific contexts, for example, in systems with little computational resources.

Author Response

Thank you for giving us the opportunity to submit a revised manuscript titled “How Does Neural Network  Model Capacity Affect Photovoltaic Power Prediction? A Study Case” to Journal Sensors. 

We appreciate the time and effort that the editors and reviewers have taken in reviewing our manuscript and providing valuable feedback. We are grateful to the reviewers for their insightful comments, which have helped us improve the manuscript significantly. 

We agree with almost all their comments and have revised the manuscript to reflect the suggested changes. We also included a point-to-point response to the reviewers, redid the abstract to avoid overlapping with the text, and revised the references to leave only the most relevant ones.

Please find below the reviewers’ comments, followed by our responses to each comment. 

Reviewer 1

Reviewer 1 point 1. This paper compared the performance of MLP (Multilayer Perceptron), Recurrent Neural Networks (RNNs), and LSTMs for the task mentioned above. Each iteration of the forecast uses the last 120 minutes of data collected from the photovoltaic (PV) system (power, irradiation, and PV cell temperature), measured throughout 2019 to mid-2022 in the city of Maceió (Brazil), to predict the next 5-minutes of PV instant power. Hyperparameters were fine-tuned using an optimization approach based on Bayesian inference to promote a fair comparison. Results showed that the MLP has satisfactory performance, requiring much less time to train and forecast. However, if well-tuned, the recurrent neural networks still present the best results. Such results indicate that MLP models can be a better option when dealing with a short-term forecast in specific contexts, for example, in systems with little computational resources.

Response 1: The authors appreciate the feedback.

Thank you again for the interest in our manuscript. We look forward to hearing from you in due time regarding our submission and to respond to any further questions and comments you may have. 

Best regards, 

Carlos Andrade

Computation Institute, Federal University of Alagoas - UFAL Maceió, Alagoas, Brazil 

Email Address: [email protected]

Reviewer 2 Report

1. try to explain the advantages and disadvantages of various physical models, statistical models and artificial intelligence models
2. cite some necessary papers on LSTM models (10.3390/w14040610, 10.1016/j.egyr.2021.09.167, 10.1016/j.epsr.2022.107908)
3. l.194, Formula error
4. the performance of the computer used in the example should make the necessary presentation
5. how the hyperparameters of the LSTM model are determined? Why not consider some necessary optimization-seeking algorithms?
6. too few comparison models, the results of some necessary machine learning models should be added, such as BP, RF, GRU, etc.

Author Response

Thank you for giving us the opportunity to submit a revised manuscript titled “How Does Neural Network  Model Capacity Affect Photovoltaic Power Prediction? A Study Case” to Journal Sensors. 

We appreciate the time and effort that the editors and reviewers have taken in reviewing our manuscript and providing valuable feedback. We are grateful to the reviewers for their insightful comments, which have helped us improve the manuscript significantly. 

We agree with almost all their comments and have revised the manuscript to reflect the suggested changes. We also included a point-to-point response to the reviewers, redid the abstract to avoid overlapping with the text, and revised the references to leave only the most relevant ones.

Point 1. try to explain the advantages and disadvantages of various physical models, statistical models and artificial intelligence models

Response 1: The authors appreciate the feedback and added the suggested information in the text.

Point 2. cite some necessary papers on LSTM models (10.3390/w14040610, 10.1016/j.egyr.2021.09.167, 10.1016/j.epsr.2022.107908)

Response 2: The authors cited the suggested articles in the text. Article 10.3390/w14040610 was included as an example of variable forecasting. The other two were used to illustrate the extensive usage of sophisticated recurrent neural networks.

Point 3. l.194, Formula error

Response 3: The authors made the suggested correction by replacing the misplaced ‘e’ for an ‘and’.

Point 4. the performance of the computer used in the example should make the necessary presentation

Response 4: The authors made the suggested correction, inserting the computer configuration after the Table 3 citation.

Point 5. how the hyperparameters of the LSTM model are determined? Why not consider some necessary optimization-seeking algorithms?

Response 5: The authors appreciate the feedback. We made it more apparent in the text that the LSTM hyperparameter optimization process was the same as the other two.

Point 6. too few comparison models, the results of some necessary machine learning models should be added, such as BP, RF, GRU, etc.

Response 6: Due to the long time required, especially to optimize the hyperparameters of the neural networks, the authors decided not to include these comparisons in this text, intending to leave this possibility for future works. Therefore, we added this possibility in the conclusion section as future work.

Thank you again for your interest in our manuscript. We look forward to hearing from you regarding our submission and responding to any further questions and comments you may have. 

Best regards, 

Carlos Andrade

Computation Institute, Federal University of Alagoas - UFAL Maceió, Alagoas, Brazil 

Email Address: [email protected]

Reviewer 3 Report

This paper presents a comparative study between the MLP, RNNs, and LSTM. Corresponding accuracy and time of training are analyzed and compared. The topic is interesting. However, several comments or suggestions are presented here for authors.

1-In the Introduction, the authors state “However, the computational cost of a deep recurrent neural network with many parameters can be prohibitively expensive and take too long to train.” However, the references are required for supporting this statement.

2-The algorithms predict five minutes of power generation using variables of the past 120 minutes. This is the aspect of ultra-short term power prediction of PV systems, not the short-term forecast. The statement of “short-term forecast” should be revised.

3-In Section 2.1, the data acquisition is described. However, the total capacity of the experimental PV system under STC is unknown. Otherwise, in Table 2, the MAE, RMSE, STD are all in Watts and are difficult to understand, if the total capacity of the experimental PV system under STC is unknown.

4-In Section 2.2, before the data are analyzed, the samples acquired at night are useless and should be filtered. Otherwise, the average value in the boxplots cannot accurately present the data distribution. The networks models are not necessarily trained by successive data, since the power is not needed to be predicted at night.

5-Figure 8 only shows the prediction results in a cloudy day. The results under sunny and partial cloudy days should be presented as well.

Author Response

Thank you for giving us the opportunity to submit a revised manuscript titled “How Does Neural Network  Model Capacity Affect Photovoltaic Power Prediction? A Study Case” to Journal Sensors. 

We appreciate the time and effort that the editors and reviewers have taken in reviewing our manuscript and providing valuable feedback. We are grateful to the reviewers for their insightful comments, which have helped us improve the manuscript significantly. 

We agree with almost all their comments and have revised the manuscript to reflect the suggested changes. We also included a point-to-point response to the reviewers, redid the abstract to avoid overlapping with the text, and revised the references to leave only the most relevant ones.

Reviewer: 

This paper presents a comparative study between the MLP, RNNs, and LSTM. Corresponding accuracy and time of training are analyzed and compared. The topic is interesting. However, several comments or suggestions are presented here for authors.

Point 1. In the Introduction, the authors state “However, the computational cost of a deep recurrent neural network with many parameters can be prohibitively expensive and take too long to train.” However, the references are required for supporting this statement.

Response 1: The authors appreciate the feedback and made the suggested correction citing article 10.1109/ASRU.2017.8268926 to support the statement.

Point 2. The algorithms predict five minutes of power generation using variables of the past 120 minutes. This is the aspect of ultra-short term power prediction of PV systems, not the short-term forecast. The statement of “short-term forecast” should be revised.

Response 2: The authors appreciate the feedback and, based on other articles, changed every instance of the term "short-term" to "very short-term" in the text.

Point 3. In Section 2.1, the data acquisition is described. However, the total capacity of the experimental PV system under STC is unknown. Otherwise, in Table 2, the MAE, RMSE, STD are all in Watts and are difficult to understand, if the total capacity of the experimental PV system under STC is unknown.

Response 3: The authors made the suggested correction by stating the total capacity of 5 kW of the PV system under STC in the second paragraph.

Point 4. In Section 2.2, before the data are analyzed, the samples acquired at night are useless and should be filtered. Otherwise, the average value in the boxplots cannot accurately present the data distribution. The networks models are not necessarily trained by successive data, since the power is not needed to be predicted at night.

Response 4: The authors appreciate the feedback. To clarify this behavior of the data, we added a new boxplot of the PV instant power using only the daytime values. Furthermore, we updated Figures 5b and 5c  to new boxplots containing only the daytime values.

Point 5. Figure 8 only shows the prediction results in a cloudy day. The results under sunny and partial cloudy days should be presented as well.

Response 5: The authors appreciate the feedback. We added prediction examples for one sunny day and one that fit the characteristics of a partially cloudy day.

Thank you again for your interest in our manuscript. We look forward to hearing from you regarding our submission and responding to any further questions and comments you may have.  

Best regards, 

Carlos Andrade

Computation Institute, Federal University of Alagoas - UFAL Maceió, Alagoas, Brazil 

Email Address: [email protected]

Reviewer 4 Report

This manuscript studied the prediction of solar power generation using MLP, RNNs and LSTMs and compared the results to report the optimum approach. The article is quite interesting and very well organized. I have just a few suggestions before its possible publication in Sensors.

Author Response

Thank you for giving us the opportunity to submit a revised manuscript titled “How Does Neural Network  Model Capacity Affect Photovoltaic Power Prediction? A Study Case” to Journal Sensors. 

We appreciate the time and effort that the editors and reviewers have taken in reviewing our manuscript and providing valuable feedback. We are grateful to the reviewers for their insightful comments, which have helped us improve the manuscript significantly. 

We agree with almost all their comments and have revised the manuscript to reflect the suggested changes. We also included a point-to-point response to the reviewers, redid the abstract to avoid overlapping with the text, and revised the references to leave only the most relevant ones.

Reviewer:

This manuscript studied the prediction of solar power generation using MLP, RNNs and LSTMs and compared the results to report the optimum approach. The article is quite interesting and very well organized. I have just a few suggestions before its possible publication in Sensors.

Point 1. The title of the manuscript could be altered if possible. For example: “The effects of neural network model capacity on photovoltaic power prediction.” OR if still you want to keep the title as a question, you would better to change “A study case” after the question mark.

Response 1: The authors appreciate the feedback and removed “A study case” from the title.

Point 2. Line 105, “With this” could be changed to “Using the method” or “By means of that”.

Response 2: The authors made the suggested corrections.

Point 3. Table 1, “Characteristic” to “Characteristics” or “specifications”.

Response 3: The authors made the suggested corrections.

Point 4. Line 165, “1e-3” can be changed to “1x10ˆ-3” or “1/1000” to be consistent with the rest of the manuscript.

Response 4: The authors made the suggested corrections.

Point 5. Figure 7, the name of horizontal and vertical axes could be changed to be consistent with the rest of the manuscript.

Response 5: The authors made the suggested corrections about the vertical axes. We decided to keep the horizontal axis as it follows the pattern of other charts.

Point 6. I think it would be beneficial if you read and also cite some of the following articles in terms of prediction and optimization:

  • Short-term photovoltaic power forecasting using Artificial Neural Networks and an Analog Ensemble
  • Optimum model for bearing capacity of concrete-steel columns with AI technology via incorporating the algorithms of IWO and ABC
  • Machine-learning methods for integrated renewable power generation: A comparative study of artificial neural networks, support vector regression, and Gaussian Process Regression

Response 6: The authors found the articles interesting and cited the text's first and third suggested articles. The first article was used as an example of a hybrid approach. The third one was used in the conclusion section as an idea for future work, incorporating the possibilities of models for comparisons.

Thank you again for your interest in our manuscript. We look forward to hearing from you regarding our submission and responding to any further questions and comments you may have.  

Best regards, 

Carlos Andrade

Computation Institute, Federal University of Alagoas - UFAL Maceió, Alagoas, Brazil 

Email Address: [email protected] 

Round 2

Reviewer 2 Report

The author's revisions are satisfactory. I agree to accept the paper in this form.

Reviewer 3 Report

The authors have addressed the comments and revised the manuscript properly.